# Causal Associations Between Pre-Pregnancy Diabetes Mellitus and Pre-Eclampsia Risk: Insights from a Mendelian Randomization Study

**DOI:** 10.3390/healthcare13091085

**Published:** 2025-05-07

**Authors:** Xiang Ying, Quanfeng Wu, Xiaohan Li, Yan Bi, Li Gao, Shushu Yu, Xiaona Xu, Xiaotian Li, Yanlin Wang, Renyi Hua

**Affiliations:** 1Division of Fetal Medicine, Prenatal Diagnosis Department, International Peace Maternity and Child Health Hospital, Shanghai Jiao Tong University School of Medicine, Shanghai 200030, China; yxlilian@alumni.sjtu.edu.cn (X.Y.); xiaohan0509@sjtu.edu.cn (X.L.); rimuzi@sjtu.edu.cn (L.G.); dorayu@sjtu.edu.cn (S.Y.); ggappsong@sjtu.edu.cn (X.X.); 2Shanghai Key Laboratory of Embryo Original Disease, Shanghai 200030, China; byan0908@sjtu.edu.cn; 3Department of Obstetrics, Shenzhen Maternity and Child Healthcare Hospital, Southern Medical University, Shenzhen 518028, China; qf.wu@xmu.edu.cn (Q.W.); xtli555@fudan.edu.cn (X.L.); 4Department of Obstetrics, Women and Children’s Hospital, School of Medicine, Xiamen University, Xiamen 361003, China; 5Department of Obstetrics, Obstetrics and Gynecology Hospital of Fudan University, Shanghai 200011, China

**Keywords:** pre-eclampsia, diabetes mellitus, Mendelian randomization

## Abstract

Background and Objectives: Pre-eclampsia (PE) is a serious pregnancy complication defined by the onset of hypertension and multi-organ dysfunction occurring after 20 weeks of gestation. Studies have indicated the correlation between diabetes mellitus (DM) and PE, but the causal relationship remains unclear. Materials and Methods: The two-sample Mendelian randomization (MR) approach, including the inverse variance weighted random effects (IVW-RE) model and the traditional sensitivity model, was employed to assess the causal effects of pre-pregnancy type 1 diabetes (T1D) and type 2 diabetes (T2D) on PE using summary-level data obtained from genome-wide association studies. Additionally, diabetes-related factors, such as glycated hemoglobin (HbA1c) levels, fasting insulin levels, and body mass index (BMI), were evaluated for their potential causal effects on the risk of PE. Pleiotropy-robust and multivariable Mendelian randomization (MVMR) methods were further used because of the intricate associations among the traits. Insulin and metformin use was also assessed for their causal role in PE risk. Results: Our findings show that genetically predicted T1D (OR = 1.06, 95% CI: 1.03–1.09, *p* < 0.001), T2D (OR = 1.09, 95% CI: 1.04–1.14, *p* < 0.001), and BMI (OR = 1.64, 95% CI 1.49 to 1.80, *p* < 0.001) had causal effects on the incidence of PE, while the effects of HbA1c (OR = 0.77, 95% CI 0.59 to 1.02, *p* = 0.064) and fasting insulin levels (OR = 1.35, 95% CI 0.89 to 2.05, *p* = 0.153) on the occurrence of PE were not significant. The results were verified by MVMR analysis. Additionally, insulin use increased the risk of pre-eclampsia (OR = 1.11, 95% CI 1.05–1.17, *p* < 0.001). Conclusions: Our findings demonstrate a causal relationship between pre-pregnancy diabetes (DM) and obesity and the risk of PE from a genetic epidemiological perspective. Adverse maternal factors, including DM and obesity prior to pregnancy, should be considered in mechanistic studies of PE. In addition, comprehensive interventions for risk factors such as pre-pregnancy DM and obesity should be emphasized in clinical practice.

## 1. Introduction

Pre-eclampsia (PE) is a severe pregnancy complication defined by new-onset hypertension after 20 weeks of gestation, potentially leading to damage across various organs and systems. Currently, PE remains one of the main causes of morbidity and mortality for both pregnant women and perinatal infants globally [1,2]. In addition, PE is associated with an increased long-term risk of cardiovascular diseases, kidney diseases, and metabolic syndrome in the mother. Meanwhile, the offspring of women with PE may encounter risks related to neurodevelopment, cardiovascular health, and metabolism, posing significant threats to the health of mothers and infants and adversely affecting the quality of the birth population [3,4]. Currently, there remains an absence of effective curative treatments for PE. The only way to cure PE involves the delivery of the fetus and placenta [1]. The understanding of the risk factors and pathogenesis of PE remains incomplete, significantly hindering the effective improvement of prevention and treatment.

DM (diabetes mellitus) is a systemic metabolic disorder primarily characterized by hyperglycemia and has emerged as a significant challenge in the global public health domain. Studies have revealed that the global prevalence of DM exceeds 400 million individuals, with a consistently increasing incidence rate [5,6,7]. T1D is caused by the autoimmune destruction of pancreatic β-cells, leading to insulin deficiency [8]. Type 2 diabetes (T2D) is defined by insulin resistance, dysregulated blood glucose regulation, and the gradual deterioration of β-cell function [9]. Numerous studies have shown that pre-pregnancy DM significantly increases the risk of developing PE. The incidence of PE ranges among pregnant women with T1D from 9% to 20% [10,11,12,13], while in those with T2D, the incidence ranges from 7% to 14% [14]. However, the causal relationship between pre- pregnancy DM and its subtypes and the pathogenesis of PE remains incompletely understood.

Maternal DM prior to pregnancy may be causally linked to PE through intricate underlying mechanisms. Chronic hyperglycemia is a key characteristic of pre-pregnancy DM [8,9]. Hyperglycemia can contribute to PE by increasing oxidative stress and inflammatory responses, thereby impairing vascular endothelial function [15,16]. Hyperglycemia may also lead to the inadequate remodeling of spiral arteries, reducing placental blood flow and resulting in placental ischemia and dysfunction, thus contributing to the occurrence of PE [17,18]. Additionally, individuals with pre-pregnancy DM also usually have insulin resistance [19,20], which may exacerbate vascular function impairment and negatively impact placental health. Although the abovementioned mechanisms offer valuable insights, the causal relationship between pre-pregnancy DM and PE still warrants further investigation.

Traditional observational studies often encounter challenges such as residual confounding and reverse causality. MR is a statistical method designed to assess the causal relationship between two related variables. This method employs randomly assigned genetic variants as instrumental variables (IVs) for phenotypes, thereby establishing causal inferences regarding the relationship between exposure and outcome. Genetic variants are established at the time of fertilization, allowing MR to investigate causal relationships without being affected by environmental influences and reverse causality [21]. Therefore, MR provides a powerful means for exploring the complex causal relationship between pre-pregnancy DM and PE, providing valuable insights into the role of maternal factors in the pathogenesis of PE.

The primary objective of this study was to evaluate the causal relationship between maternal T1D and T2D and the risk of PE using the two-sample MR approach. Furthermore, the causal effects of various continuous exposure variables related to diabetes, including average blood glucose levels (with glycated hemoglobin (HbA1c) levels as the exposure variable), insulin signaling (with fasting insulin levels as the exposure variable), and the extent of obesity (with BMI as the exposure variable), on PE were assessed through MR analysis. This study employed multivariable Mendelian randomization (MVMR) analysis to assess the independent causal effects of DM and its associated factors (HbA1c, fasting insulin, and BMI) on PE, given the complexity of the pathophysiological mechanisms involved in DM. A comprehensive summary of this study is presented in Figure 1.

## 2. Materials and Methods

### 2.1. Study Design

Genetic IVs were selected according to the following criteria. First, a strong association exists between the instrumental variable and the exposure of interest. Second, the IVs must be independent of confounding factors that affect both the exposure and the outcome. Third, the IVs are related to the outcome only through the effects of the exposure.

Here, multiple complementary pleiotropy and robust statistical approaches were employed to enhance the robustness of the findings. Pleiotropy refers to the phenomenon whereby a single genotype influences multiple distinct phenotypes. Horizontal pleiotropy may cause genetic instruments affecting the outcomes through pathways unrelated to the exposure. This contravenes the exclusion restriction assumption, thereby introducing bias into MR analysis. Consequently, various MR methods have been developed by researchers to provide pleiotropy-robust causal estimates. These methods generally utilize different statistical techniques to attain pleiotropy and robust causal estimates. However, each method possesses distinct limitations. Consequently, adopting multiple complementary methods is advisable.

In the two-sample MR study design, the association between genetic variants and exposure variables is initially estimated in one dataset. The relationship between the same genetic variants and the outcome are subsequently evaluated in another independent dataset. This design allows the two-sample MR method to explore the causal relationships between variables that are not directly measured in different cohorts. Moreover, this method is applicable in a multivariable context. By integrating IVs for multiple relevant exposure factors, it is possible to estimate the independent causal effects of each exposure factor on the outcome [22].

### 2.2. Study Populations

Research on T1D [23] (6683 cases, 12,173 controls, 2601 affected siblings, and 69 trios of European ancestry), T2D [24] (61,714 cases and 593,952 controls of European ancestry), PE [25] (9023 cases and 259,313 controls of European ancestry), HbA1c [26] (146,806 samples of European ancestry), fasting insulin levels [26] (151,013 samples of European ancestry), and BMI [27] (681,275 samples of European ancestry) has indicated that GWASs utilizing the same ancestral background enhance the accuracy of causal inference by mitigating confounding effects associated with population stratification.

In the HbA1c and fasting insulin GWAS cohorts, participants diagnosed with T1D or T2D were excluded, as were individuals using DM treatment or those with blood glucose measurements meeting the DM diagnostic criteria (e.g., fasting blood glucose ≥ 7 mmol/L, 2 h blood glucose ≥ 11.1 mmol/L, or HbA1c ≥ 6.5%). Excluding these DM participants allowed for a clearer identification of genetic associations with HbA1c and fasting insulin, thereby mitigating potential confounding effects from DM treatment.

### 2.3. Genetic Instrument Selection

The selected genetic variants must be significantly associated with the exposure variable (*p* < 5 × 10^−8^) to ensure adherence to the correlation-based MR assumption. Independent signals were selected using a clustering method based on the 1000 Genomes Project reference panel, with r^2^ < 0.001 within a 10,000 kb window. All genetic variants must serve as sufficiently robust IVs (F > 10) to avoid weak instrument bias. Missing genetic variants should be replaced with appropriate proxy variants (R^2^ > 0.8).

### 2.4. Statistical Analyses

The inverse variance weighted random effects (IVW-RE) method [28] was employed to estimate the overall effects of each exposure on the outcome utilizing the R software (version 4.4.1) package. MR-Egger [29] and weighted median (MR-WM) [30] methods were used to assess pleiotropy-robust causal estimates for sensitivity analysis. The MR-Egger method estimated causal effects through meta-regression, adjusting for directional pleiotropy, while the MR-WM method permitted pleiotropic effects in up to 50% of the IVs. Radial MR identified and eliminated outliers by identifying variants that significantly contribute to heterogeneity, based on their weights and distance from the consensus causal estimates. These variants may introduce bias into causal estimates [31]. MR-PRESSO [32] and MR-Lasso [33] are extensions of the IVW method designed to identify and eliminate potentially pleiotropic genetic instruments prior to the computation of causal estimates. MR-PRESSO identified probable pleiotropic instruments by analyzing variant contributions to heterogeneity in the IVW meta-analysis of variant-specific MR estimates, while MR-Lasso detected invalid instruments by estimating genetic effects on the outcome that by passed the exposure [33]. Leave-one-out MR was conducted for all primary analyses to assess whether individual variants were independently influencing or distorting overall causal estimates. All univariable MR methods, except for MR-PRESSO, have corresponding multivariable versions utilized in MVMR (MVMR-IVW, MVMR-Median, MVMR-Egger, and MVMR-Lasso). In MVMR, the identical genetic variants employed for each exposure in the univariable analysis were integrated into a multivariable genetic instrument [34,35]. A subsequent variant clumping step was implemented to eliminate duplicate variants linked to multiple exposure variables.

## 3. Results

### 3.1. Selection of Genetic Instruments

A total of 30 genetic variants were selected as IVs for T1D and 110 genetic variants as IVs for T2D. A total of 65, 32, and 468 genetic variants were utilized to assess HbA1c levels, fasting insulin levels, and BMI, respectively. All genetic variants selected for these exposures were evaluated in a relevant genome-wide association study (GWAS) associated with PE.

### 3.2. Causal Association of DM Types with PE

MR analysis revealed that genetically predicted T1D exerted a significant causal influence on the incidence of PE (IVW-RE, OR = 1.06, 95% CI 1.03–1.09, *p* < 0.001) (Figure 2). Similarly, genetically predicted T2D demonstrated a significant causal relationship with PE (IVW-RE, OR = 1.09, 95% CI 1.04–1.14, *p* < 0.001) (Figure 2). Heterogeneity analysis indicated that neither T1D (Q = 21.94, *p* = 0.823, I^2^ = 0%) nor T2D (Q = 104.87, *p* = 0.594, I^2^ = 0%) exhibited significant heterogeneity in their associations with PE. Meanwhile, the results of the pleiotropy-robust analysis were consistent with those obtained by the IVW-RE, along with weighted median (T1D: OR = 1.06, 95% CI 1.01–1.10, *p* = 0.010; T2D: OR = 1.12, 95% CI 1.04–1.22, *p* = 0.003) and MR-Lasso (T1D: OR = 1.06, 95% CI 1.03–1.09, *p* < 0.001; T2D: OR = 1.08, 95% CI 1.04–1.14, *p* < 0.001). The results of MR-PRESSO analysis further supported these findings (T1D: OR = 1.06, 95% CI 1.03–1.09, *p* < 0.001; T2D: OR = 1.09, 95% CI 1.04–1.14, *p* < 0.001). Leave-one-out MR analysis demonstrated that no genetic variants were identified that could independently influence or distort the overall causal estimates for T1D on PE (Appendix A) and T2D on PE (Appendix A).

### 3.3. Causal Effect Assessment via Continuous Exposure and MVMR

No evidence indicated that HbA1c was involved in the causal pathway of PE (IVW-RE, OR = 0.77, 95% CI 0.59 to 1.02, *p* = 0.064) (Figure 3). In addition, no significant heterogeneity was detected (Q = 61.61, *p* = 0.56, I^2^ = 0%). The estimates obtained from pleiotropy-robust analyses were consistent with the findings of the IVW-RE approach. The leave-one-out MR analysis did not identify any genetic variants that could independently drive or distort the overall causal estimates (Appendix A). In MVMR analysis, after adjusting for HbA1c, both T1D (MV-IVW, OR = 1.065, 95% CI 1.02–1.11, *p* = 0.003) and T2D (MV-IVW, OR = 1.14, 95% CI 1.08–1.21, *p* < 0.001) exerted significant independent effects on the risk of PE. Interestingly, HbA1c showed no significant effect when analyzed with T1D (MV-IVW, OR = 0.78, 95% CI 0.35–1.73, *p* = 0.541) but demonstrated a significant protective effect when analyzed with T2D (MV-IVW, OR = 0.59, 95% CI 0.40–0.86, *p* = 0.007) (Figure 4 and Figure 5). The aforementioned findings were generally consistent with the results obtained by pleiotropy-robust methods, including the MVMR-Median, MVMR-Egger, and MVMR-Lasso methods, all indicating comparable patterns of association.

No causal association was observed between fasting insulin levels and PE (IVW-RE, OR = 1.35, 95% CI 0.89 to 2.05, *p* = 0.153) (Figure 3). No significant heterogeneity was detected in this study (Q = 36.56, *p* = 0.27, I^2^ = 12.47%). In addition, the results estimated by pleiotropy-robust analyses (weighted median: OR = 1.36, 95% CI 0.79 to 2.35, *p* = 0.268; MR-Egger: OR = 0.68, 95% CI 0.18 to 2.63, *p* = 0.575) were consistent with those obtained through the inverse variance weighted random effects (IVW-RE) method. In the leave-one-out MR analysis, no genetic variants were identified that could independently influence or bias the overall causal estimates (Appendix A). The results of the MVMR analysis demonstrated that, following adjustments for fasting insulin levels, T1D (MVMR-IVW, OR = 1.06, 95% CI 1.02 to 1.10, *p* = 0.002) and T2D (MVMR-IVW, OR = 1.11, 95% CI 1.05 to 1.17, *p* < 0.001) exerted significant independent effects on the risk of PE. The impact of fasting insulin differs across various models: in the T1D analysis, certain methods indicated a possible protective effect (MVMR-Median: OR = 0.79, 95% CI 0.65 to 0.97, *p* = 0.024; MVMR-Lasso: OR = 0.78, 95% CI 0.68 to 0.90, *p* < 0.001); conversely, the primary MVMR-IVW analysis in conjunction with T2D revealed no significant association (OR = 0.97, 95% CI 0.88 to 1.08, *p* = 0.586), while the MVMR-Lasso method suggested a moderate protective effect (OR = 0.91, 95% CI 0.833 to 0.99, *p* = 0.025) (Figure 4 and Figure 5). The abovementioned findings were generally consistent with the results obtained by pleiotropy-robust methods, including the MVMR-Median, MVMR-Egger, and MVMR-Lasso methods, all demonstrating comparable patterns of association.

BMI demonstrated a strong causal relationship with PE (IVW-RE, OR = 1.64, 95% CI 1.49 to 1.80, *p* < 0.001) (Figure 3). No significant heterogeneity was observed (Q = 368.99, *p* > 0.99, I^2^ = 0%), and the leave-one-out MR analysis did not identify any genetic variants that could independently influence or distort the overall causal estimates (Appendix A). Both the MR-PRESSO (OR = 1.64, 95% CI 1.50 to 1.79, *p* < 0.001) and weighted median (OR = 1.70, 95% CI 1.47 to 1.98, *p* = 2.05 × 10^−12^) methods corroborated the findings of the IVW-RE analysis. The MR-Egger analysis indicated consistent results with substantial effect sizes (OR = 2.05, 95% CI 1.54 to 2.74, *p* < 0.001), suggesting the robustness of the causal associations obtained from different MR approaches.

MVMR showed that both DM types and BMI exerted significant independent effects. Upon adjusting for BMI, T1D (MVMR-IVW, OR = 1.07, 95% CI 1.03–1.11, *p* < 0.001) and T2D (MVMR-IVW, OR = 1.10, 95% CI 1.05–1.16, *p* < 0.001) remained significantly associated with PE. Notably, BMI demonstrated significant independent effects in both models: T1D (MVMR-IVW, OR = 1.60, 95% CI 1.20–2.13, *p* = 0.002) and a stronger effect for T2D (MVMR-IVW, OR = 1.49, 95% CI 1.31–1.70, *p* < 0.001) (Figure 4 and Figure 5). These findings were consistently supported by all pleiotropy-robust approaches, including MVMR-Median, MVMR-Egger, and MVMR-Lasso, demonstrating similar or stronger associations.

### 3.4. Association Between Diabetes Treatment and PE

Using data from the IEU OpenGWAS project (Dataset: finn-b-KELA_DIAB_INSUL), we conducted a Mendelian randomization analysis with the selecting 33 IVs that met our quality criteria (Appendix A). Our analysis revealed a significant causal relationship between insulin use and the PE risk. The primary analytical method, IVW-RE, demonstrated that insulin use significantly increased the risk of pre-eclampsia (OR = 1.11, 95% CI 1.05–1.17, *p* < 0.001) (Appendix A). Other sensitivity analysis methods also supported this finding, including weighted median (OR = 1.17, 95% CI 1.08–1.26, *p* < 0.001), MR-Egger (OR = 1.15, 95% CI 1.06–1.25, *p* = 0.001), MR-Lasso (OR = 1.11, 95% CI 1.05–1.17, *p* < 0.001), and MR-PRESSO (OR = 1.11, 95% CI 1.06–1.16, *p* < 0.001). No significant heterogeneity was detected (Q = 22.38, *p* = 0.897, I^2^ = 0%).

We further investigated the causal effects of metformin use on PE risk using 30 IVs obtained from GWAS data [34] (Appendix A). The IVW-RE analysis suggested a modest, non-significant effect of metformin on pre-eclampsia risk (OR = 1.04, 95% CI 0.99–1.09, *p* = 0.140) (Appendix A). While most sensitivity analyses showed consistent non-significant results, including weighted median (OR = 1.07, 95% CI 0.99–1.15, *p* = 0.081), MR-Lasso (OR = 1.04, 95% CI 0.99–1.09, *p* = 0.140), and MR-PRESSO (OR = 1.04, 95% CI 0.99–1.09, *p* = 0.150), the MR-Egger method indicated a significant association (OR = 1.15, 95% CI 1.02–1.28, *p* = 0.018). Heterogeneity analysis showed no significant heterogeneity in the metformin analysis (Q = 29.76, *p* = 0.426, I^2^ = 2.55%), suggesting consistent effect estimates across the IVs.

## 4. Discussion

Our MR study demonstrated that pre-pregnancy DM, including T1D and T2D, as well as elevated BMI, was causally associated with the risk of PE. Interestingly, pre-pregnancy DM-related factors, such as HbA1c and fasting insulin levels, were unlikely to influence the causal pathway of PE.

The causal link between pre-pregnancy DM and PE supports previous observational findings and extends our understanding of this relationship [36,37]. Previous studies have shown that women with pre-pregnancy DM who develop PE during pregnancy already exhibited maternal vascular dysfunction in the early stages of pregnancy [38]. From the perspective of the rationality of biological mechanisms, patients with DM often present with pathological states such as insulin resistance, systemic inflammation, and endothelial dysfunction [39]. These pre-existing conditions may worsen during pregnancy, substantially increasing PE risk. Future research should focus more on personalized risk assessment and intervention strategies for PE in the diabetic population, developing more precise prevention and treatment strategies for pre-pregnancy diabetic patients with varying types, disease courses, and complications.

Furthermore, this study conducted a comprehensive assessment of the causal pathways between DM, including T1D and T2D, and PE. HbA1c has been demonstrated to outperform other diabetes-related IVs in terms of potency and variation explanation. In addition, it can elucidate potential interaction mechanisms [40]. Therefore, we incorporated the genetic IVs of HbA1c, fasting insulin, and BMI, along with DM, as exposure factors for subsequent analysis. However, the null effect of HbA1c on PE contradicted the relationships and mechanisms suggested in previous research, which indicated an association between elevated HbA1c levels and an increased risk of PE. HbA1c serves as an indicator reflecting the average blood glucose level over the past 2 to 3 months. Numerous previous studies have shown that an elevated maternal HbA1c level before or during early pregnancy was significantly associated with an increased risk of PE. The mechanisms involved the induction of endothelial dysfunction, systemic inflammatory response, and the disruption of placental development and function caused by high HbA1c levels [41,42,43,44]. We hypothesized that the association identified in observational studies may not be directly linked to HbA1c but rather to other concurrent risk factors, such as obesity. In addition, high levels of HbA1c indicate a lack of excellent DM control and high blood glucose levels, while uncontrolled DM may contribute to PE. Future research needs to employ more methods to evaluate the interactions between HbA1c and other risk factors, thereby elucidating the relationship between HbA1c and the occurrence of PE. 

Similarly, we found no causal relationship between pre-pregnancy fasting insulin levels and the risk of PE, contrasting with the results from prior observational studies [45,46]. Confounding factors in observational studies may lead to biases in the results. This study employed the MR approach for causal inference, effectively addressing common confounding and reverse causality challenges present in observational studies. In addition, insulin resistance may influence PE through indirect pathways involving oxidative stress, inflammatory, and endothelial dysfunction. Moreover, fasting insulin levels may not be the optimal indicator for assessing insulin resistance. Consequently, the complex relationship between insulin resistance and PE warrants further investigation.

Ultimately, both univariable and multivariable MR analyses demonstrated that pre-pregnancy BMI has a significant independent causal effect on the risk of PE. This finding is consistent with the established positive correlation between BMI and PE revealed in various prior observational studies [47,48,49,50] while also offering enhanced evidence for causal inference. Interestingly, the analysis of the MVMR results regarding the independent effects of DM subtypes and BMI found that, even after adjusting for BMI, T1D and T2D remain significantly positively correlated with PE incidence. This suggests that DM may contribute to the development of PE through mechanisms independent of BMI. In all models, BMI showed a significant independent effect, indicating that pre-pregnancy overweight or obesity is a major risk factor for PE and may further increase this risk in diabetic patients. Nonetheless, the intricate factors affecting BMI make the mechanisms through which BMI influences the onset of PE ambiguous. Interestingly, insulin use showed a significant positive association with PE risk, while metformin did not, indicating differential clinical implications for these medications during pregnancy preparation and gestation.

This study employed multiple complementary MR methods. These methods effectively mitigated confounding factors and reverse causality issues while confirming result robustness through sensitivity analyses [51]. The application of binary exposure variables, such as the diagnosis of T2D based on a glycated hemoglobin (HbA1c) threshold of ≥6.5%, may result in confounding [48]. To address this limitation, we performed additional two-sample MR analyses, using continuous variables (such as HbA1c, fasting insulin levels, and body mass index (BMI)) as exposure factors. Moreover, we also performed MVMR analyses to evaluate the independent effects of related exposure variables (T1D, T2D, HbA1c, fasting insulin levels, and BMI). Consequently, this study comprehensively investigated the potential causal pathways between pre-pregnancy DM and the risk of PE, addressing the limitations of traditional MR methods.

This study has some limitations. Although two-sample MR can provide valuable etiological insights, its estimates rely on summary data and usually only reflect only the overall effect. This method may miss complex genetic factors specific to disease subtypes, stages, or gene–environment interactions. Defining genetic IVs based on disease status may be considered an oversimplification of disease effects. In addition, sample racial background and environmental factors may limit result generalizability. While MR offers robust causality assessment, comprehensive interpretation requires additional evidence types according to current guidelines [52]. Future research should further validate these causal associations in larger multi-center cohorts and investigate their underlying mechanisms through basic experiments.

## 5. Conclusions

In this comprehensive Mendelian randomization analysis, we established causal relationships between pre-pregnancy diabetes mellitus and pre-eclampsia risk. Both pre-pregnancy T1D and T2D independently increased pre-eclampsia risk, with pre-pregnancy BMI demonstrating a separate significant causal influence. Conversely, pre-pregnancy HbA1c and fasting insulin levels did not demonstrate a definitive role in the causal pathway, suggesting complex underlying mechanisms. Future research should develop larger, multi-center, and multi-ethnic clinical cohorts, or conduct clinical trials to validate the findings, in order to achieve more robust and comprehensive conclusions, particularly regarding the relationship between pre-pregnancy HbA1c levels and insulin resistance and PE risk. Moreover, future research on the mechanisms of PE should consider the impact of pre-pregnancy diabetes and explore the potential therapeutic effects of anti-diabetic medications in preventing and treating PE. Clinically, these findings underscore the importance of developing more targeted individualized risk assessment models and intervention strategies for the high-risk population of pre-pregnancy DM patients.

## Figures and Tables

**Figure 1 healthcare-13-01085-f001:**
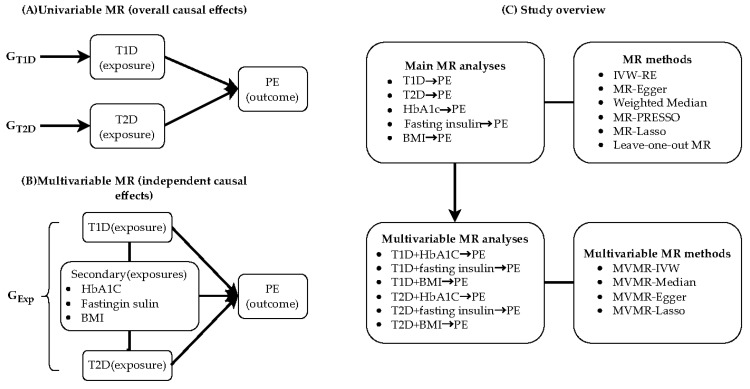
Overview of study design and analytical approaches. (**A**) Univariable MR analysis showing overall causal effects, demonstrating use of genetic IVs (GT1D and GT2D) to study causal relationships between T1D, T2D, and PE. (**B**) Multivariable MR analysis showing independent causal effects. Genetic variants associated with multiple correlated exposures (GExp) (including T1D, T2D, HbA1c, fasting insulin, and BMI) are integrated to test direct causal effects of each exposure on PE. (**C**) Study framework overview, including main MR analyses (testing relationships between T1D, T2D, HbA1c, fasting insulin, BMI, and PE), multivariable MR analyses (examining effects of T1D/T2D combined with other metabolic indicators on PE), and various MR methods employed (such as IVW-RE, MR-Egger, weighted median method, etc.).

**Figure 2 healthcare-13-01085-f002:**
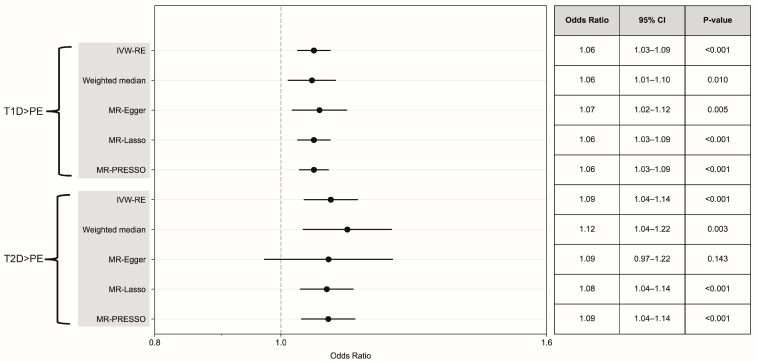
Forest plot and summary table of causal estimates from bidirectional MR analyses testing for potential causal relationships between DM (T1D and T2D) and PE. Error bars show 95% CI for overall estimates (OR) from each MR method (left). IVW-RE, inverse variance weighted random effects.

**Figure 3 healthcare-13-01085-f003:**
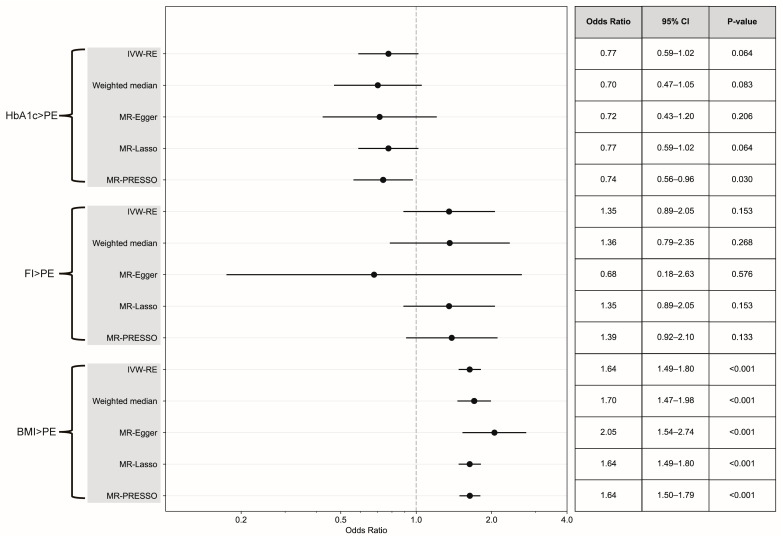
Forest plot and summary table of causal estimates from MR analyses testing for potential causal effects of continuous DM-associated variables (glycated HbA1c, fasting insulin, and BMI) on PE. Error bars show 95% CI for overall estimates (OR) from each MR method (left). IVW-RE, inverse variance weighted random effects.

**Figure 4 healthcare-13-01085-f004:**
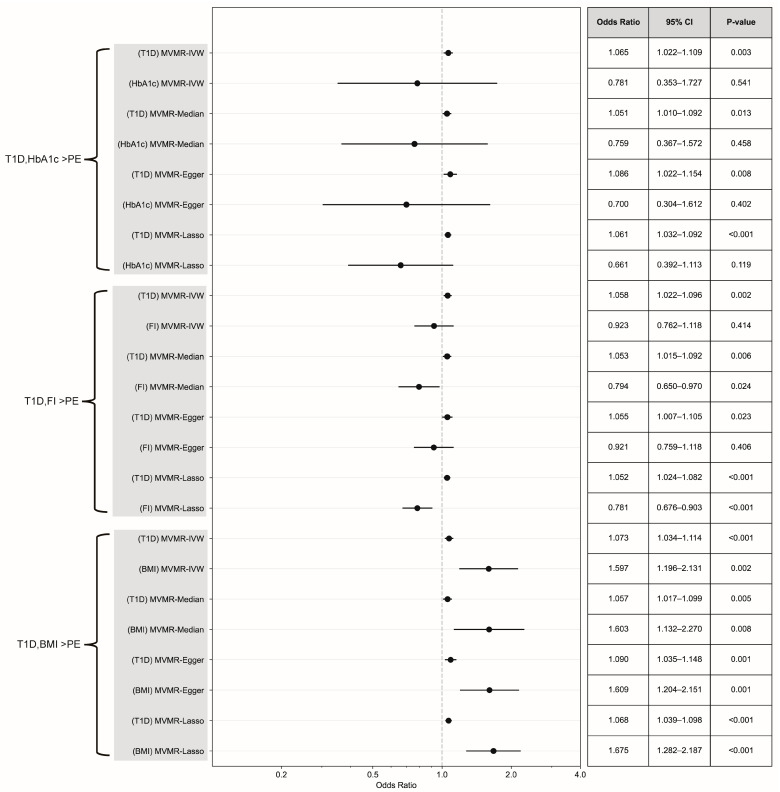
Forest plot and summary table of causal estimates from MVMR analyses testing for direct causal effects of T1D and continuous DM-associated variables (HbA1c, fasting insulin, and BMI) on PE. Error bars show 95% CI for overall estimates (OR) from each MR method (left).

**Figure 5 healthcare-13-01085-f005:**
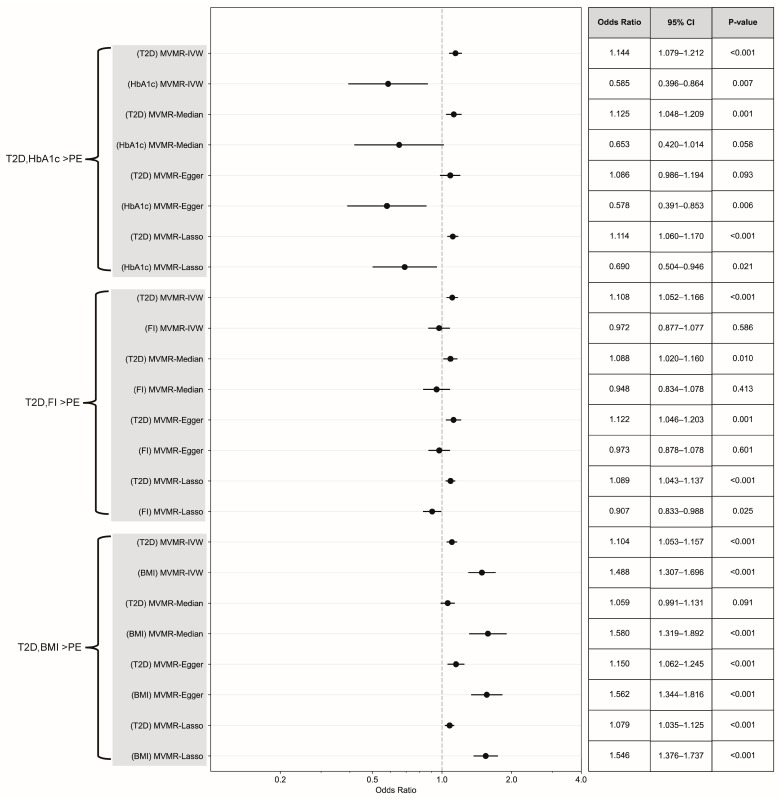
Forest plot and summary table of causal estimates from MVMR analyses testing for direct causal effects of T2D and continuous DM-associated variables (HbA1c, fasting insulin, and BMI) on PE. Error bars show 95% CI for overall estimates (OR) from each MR method (left).

## Data Availability

The data presented in this study are available on request from the corresponding author.

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
