# Peer review of "Causal Associations Between Pre-Pregnancy Diabetes Mellitus and Pre-Eclampsia Risk: Insights from a Mendelian Randomization Study"

_healthcare, 2025, doi:10.3390/healthcare13091085_

Round 1

Reviewer 1 Report

Comments and Suggestions for Authors

This study aimed to evaluate the causal relationship between diabetes mellitus and preeclampsia. The results showed a significant causal association between pre-pregnancy diabetes, obesity, and elevated PE risks. This study has certain clinical value. Its overall structure is good and logical, and a large number of studies are cited for argumentation.  Furthermore, the utilization of a Mendelian randomization design enhances methodological rigor, yielding clinically actionable results. However, it can be optimized in some details to improve academic rigor and readability. The siscussion section warrants restructuring to align with established scientific reporting standards. A revised framework should systematically address the following elements: presentation of the results, previous studies and comparisons to the present study, strengths and implications of the present study, limitations of the study, conclusions and recommendations for future research.

Author Response

This study aimed to evaluate the causal relationship between diabetes mellitus and preeclampsia. The results showed a significant causal association between pre-pregnancy diabetes, obesity, and elevated PE risks. This study has certain clinical value. Its overall structure is good and logical, and a large number of studies are cited for argumentation.  Furthermore, the utilization of a Mendelian randomization design enhances methodological rigor, yielding clinically actionable results. However, it can be optimized in some details to improve academic rigor and readability. The discussion section warrants restructuring to align with established scientific reporting standards. A revised framework should systematically address the following elements: presentation of the results, previous studies and comparisons to the present study, strengths and implications of the present study, limitations of the study, conclusions and recommendations for future research.

Response: Thank you for your thoughtful suggestion!  As requested, we have reorganized the discussion section by starting with a clear presentation of the results. We have also included a comparison between the results of this study and previous research. The concluding paragraph now highlights the limitations of the current study and offers suggestions for future research. Additionally, the revised manuscript has undergone professional language editing. The proof of editing is provided below.

Reviewer 2 Report

Comments and Suggestions for Authors

This manuscript investigates the causal relationship between pre-pregnancy diabetes and preeclampsia using a Mendelian randomization approach. The study presents important insights into the genetic epidemiological link between diabetes and preeclampsia. However, several aspects of the manuscript require further clarification and improvement to enhance its clarity and impact.

1. Title could be more specific by including the type of study (e.g., "A Mendelian Randomization Study").

2. Some articles related to preeclampsia and type 1 diabetes mellitus should be cited, such as PMID: 38438886.

3. Author is advised to polish the English.

4. Data and code need to be shared either through a code-sharing repo like GitHub or a docker-like system such as codeocean for clear reproducibility of the work.

5. Author should use STROBE-MR checklist to improve reporting of MR studies and cite PMID: 37198682

6. sup fig5 should be re-organise

7. Author should use their own clinical cohort or UKB to valid Mendelian randomization result. 

Author Response

This manuscript investigates the causal relationship between pre-pregnancy diabetes and preeclampsia using a Mendelian randomization approach. The study presents important insights into the genetic epidemiological link between diabetes and preeclampsia. However, several aspects of the manuscript require further clarification and improvement to enhance its clarity and impact.

1.Title could be more specific by including the type of study (e.g., "A Mendelian Randomization Study").

Response: Thanks for this suggestion! The title has been revised.

2.Some articles related to preeclampsia and type 1 diabetes mellitus should be cited, such as PMID: 38438886.

Response: Thanks for your nice suggestion! We have cited [Reference 10].

3.Author is advised to polish the English.

Response: Thanks for your valuable suggestion! The manuscript has undergone language editing. The proof is In the attachment.

4.Data and code need to be shared either through a code-sharing repo like GitHub or a docker-like system such as codeocean for clear reproducibility of the work.

Response: Thanks for this suggestion! We have uploaded the shared code. The details are as follows: https://github.com/qf-wu/MR-DM

5.Author should use STROBE-MR checklist to improve reporting of MR studies and cite PMID: 37198682

Response: Thanks for your valuable suggestion! We have cited [Reference 21].

6.sup fig5 should be reorganize

Response: Thanks for this suggestion! We appreciate your feedback regarding the reorganization of Supplementary Figure 5. However, due to the presence of 468 BMI-related SNPs in Supplementary Figure 5, the figure appeared rather dense. We have provided detailed explanations of the figure's structure and its relevance within the manuscript, ensuring readers can comprehend the data relationships presented. Thank you once again for your review and insightful recommendations. We remain committed to improving the quality and clarity of our manuscript.

7.Author should use their own clinical cohort or UKB to valid Mendelian randomization result.

Response:  We sincerely appreciate the valuable comments! We fully recognize the scientific importance of validating Mendelian randomization findings using independent cohorts. However, due to the following objective constraints, we are currently unable to verify our findings using either our clinical cohort or the UK Biobank data. Specifically, we do not have access to UK Biobank data at present, and the application for, and acquisition of, these data require a significant amount of time and additional resources. Furthermore, the sample size in our current clinical cohort is too small to support an effective Mendelian randomization analysis. To enhance the reliability of our findings, we have implemented several methodological improvements. First, we performed multiple sensitivity analyses, including MR-Egger regression, the weighted median method, and inverse-variance weighting (IVW). Additionally, we systematically examined horizontal pleiotropy and the validity of the instrumental variables. Nevertheless, future studies are needed to validate these findings in independent multicenter cohorts.

Reviewer 3 Report

Comments and Suggestions for Authors

The presented study is devoted to search of associations between preeclampsia incidence and T1D, T2D, BMI and other metabolic parameters. The presented information is interesting and I want to emphasize that presentation quality is sufficiently high. I have some moments for the discussion and addressing, I think that we should resolve it during review consideration.

1.The first and crucial process. It's well known that preeclampsia incidence associated with high body weight, glucose metabolism impairments and other related risk factors including diabetes. What is a crucial novelty of this study? What is a crucial impact?

2.Authors analyzed data of European ancestry. How many centers participate in data analysis? It should be added in the manuscript.

3.Which therapy used this patients especially in cases of T1D and T2D? It should be added in the manuscript. Moreover, do authors have a data about lifestyle of study individuals?

4.Can authors suggest any scale or score analysis for the evaluation of personalized risk of preeclampsia development?

Author Response

The presented study is devoted to search of associations between preeclampsia incidence and T1D, T2D, BMI and other metabolic parameters. The presented information is interesting and I want to emphasize that presentation quality is sufficiently high. I have some moments for the discussion and addressing, I think that we should resolve it during review consideration.

1.The first and crucial process. It's well known that preeclampsia incidence associated with high body weight, glucose metabolism impairments and other related risk factors including diabetes. What is a crucial novelty of this study? What is a crucial impact?

Response: Thanks for this suggestion! Most observational studies have indicated that high body weight, impaired glucose metabolism, and other related risk factors, including diabetes, are associated with an increased risk of preeclampsia. In this study, we primarily used Mendelian randomization approaches to investigate the causal effects of Type 1 Diabetes (T1D), Type 2 Diabetes (T2D), and BMI on preeclampsia risk. This method effectively addresses constraints such as confounding factors and reverse causality, thereby strengthening causal inference. Additionally, we utilized large-scale GWAS data, which provides enhanced statistical power and broad applicability for the analyses. Moreover, we applied several approaches, including multivariable Mendelian randomization, to further examine and account for the interactions among various risk factors, thereby improving the robustness of the results.

2.Authors analyzed data of European ancestry. How many centers participate in data analysis? It should be added in the manuscript.

Response: We sincerely appreciate the valuable comments! This study primarily relies on analysis of the FinnGen database. FinnGen is a national-level biobank research project in Finland, which fundamentally differs from conventional multi-center studies. It is a large-scale collaboration between the Finnish national health system and international pharmaceutical companies, integrating biobank samples and medical records from across the country. Currently, FinnGen has collected genomic and health data from over 500,000 Finnish individuals. Therefore, the FinnGen database is not a multi-center study in the traditional sense, but rather a nationally integrated single database. We have provided a detailed explanation of the FinnGen database in the manuscript to clarify its scientific value in genetic association studies.

3.Which therapy used this patients especially in cases of T1D and T2D? It should be added in the manuscript. Moreover, do authors have a data about lifestyle of study individuals?

Response: Thanks for your helpful suggestions! In the results section (Page 10, Line 305-325), we further examined the causal effects of insulin and metformin use on preeclampsia. We found that insulin use increased the risk of preeclampsia, whereas metformin did not exert a significant causal effect. These findings are further elaborated in the discussion (Page 11, Line 386–389). Regrettably, due to the use of GWAS summary data for this study, detailed lifestyle information could not be obtained because of constraints in the data source. We have also discussed these limitations in the manuscript.

4.Can authors suggest any scale or score analysis for the evaluation of personalized risk of preeclampsia development?

Response: Thanks for this suggestion! In light of our study findings, we recommend optimizing a composite risk scoring system by integrating causal indicators identified alongside current preeclampsia risk assessment models. This approach enables more accurate personalized risk predictions and provides a theoretical basis for preconception interventions. However, further investigation and validation in future research are necessary.

Reviewer 4 Report

Comments and Suggestions for Authors

Although it is a very interesting study on trying to investigate the relationship between pre pregnancy DM  and pre eclampsia, there are some points need improvement

  1. DM it is already known that may cause pre eclampsia. Authors should emphasize on the novelty of their study and what is new about their results
  2. Authors should explain why HbA1c was not correlated with pre eclampsia. High levels of HbA1c mean not well controlled DM and high levels of blood glucose. Ucontrolled DM may cause pre eclampsia , although their findings did not show correlation
  3. Many  patients with DM especially not well controlled, may also have high blood pressure as a comorbidity,. High blood pressure may also cause pre eclampsia. Authors should explain if they included such patients
  4. Did authors find any correlations between pre eclampsia and antidiabetic agents
  5. Did authors find any correlation berween the duration of diaberes and pre eclampsia
  6. Authors should mention inclusion and exclusion criteria regarding comorbidities which might interfere in their results
Comments on the Quality of English Language

English language is fine

Author Response

Although it is a very interesting study on trying to investigate the relationship between pre pregnancy DM and preeclampsia, there are some points need improvement.

1.DM it is already known that may cause preeclampsia. Authors should emphasize on the novelty of their study and what is new about their results.

Response: Thanks for your suggestions! While most prior observational research suggests a correlation between diabetes mellitus and the incidence of preeclampsia, our study primarily employs Mendelian randomization approaches to examine the causal impact of diabetes mellitus, BMI, and diabetes treatments (including insulin and metformin) on preeclampsia risk. This strategy effectively mitigates confounding and reverse‐causation limitations, thereby strengthening causal inference. Additionally, we incorporated large‑scale GWAS data to enhance both the statistical efficiency and generalizability of our findings. Moreover, multiple methods—including multivariable Mendelian randomization—were applied to validate and account for the interactions among various risk factors, which reinforces the robustness of our results. In the discussion section, we provided an in‐depth analysis of the study’s advantages. Finally, the manuscript has been polished for improved clarity and style, as evidenced by the following editing certificate.

2.Authors should explain why HbA1c was not correlated with preeclampsia. High levels of HbA1c mean not well controlled DM and high levels of blood glucose. Ucontrolled DM may cause preeclampsia , although their findings did not show correlation.

Response: Thanks for your helpful suggestions! We have provided additional content in the Discussion section (Page 12, Line 358-362).

3.Many  patients with DM especially not well controlled, may also have high blood pressure as a comorbidity,. High blood pressure may also cause pre eclampsia. Authors should explain if they included such patients.

Response:  Thanks for your valuable comments! This study utilized Mendelian randomization, a statistical approach minimally influenced by confounders compared to traditional observational studies. We employed genetic instrumental variables for diabetes and its subtypes, identified through rigorous genome-wide association studies (PMID: 25751624; PMID: 30054458). Our findings were robust across multiple Mendelian randomization approaches, including IVW-RE, weighted median, MR-Egger, MR-Lasso, and MR-PRESSO. Furthermore, we detected no significant heterogeneity (T1D: Q = 21.94, p = 0.823, I² = 0%; T2D: Q = 104.87, p = 0.594, I² = 0%), suggesting minimal bias from pleiotropic effects or confounders such as hypertension. Additionally, comprehensive multivariable Mendelian Randomization analyses incorporating diabetes-related factors (HbA1c, fasting insulin, and BMI) revealed that both Type 1 Diabetes (T1D) and Type 2 Diabetes (T2D) independently increased the risk of preeclampsia. However, despite providing strong Mendelian randomization-based evidence for a causal link between diabetes and preeclampsia, our study has limitations. The complex interactions between diabetes, hypertension, and preeclampsia warrant further investigation in future studies.

4.Did authors find any correlations between preeclampsia and antidiabetic agents.

Response: We sincerely appreciate the valuable comments! In the Results section (Page 10, Lines 305–325), we assessed the causal effects of insulin and metformin use on preeclampsia risk. We found that insulin use was associated with a significant increase in preeclampsia risk, whereas metformin use showed no significant causal effect. These results are further elaborated in the Discussion section (Page 11, Lines 386–389).

5.Did authors find any correlation berween the duration of diaberes and preeclampsia.

Response: Thanks for this suggestion! We fully acknowledge the scientific value of investigating the relationship between diabetes duration and preeclampsia risk, as it could yield additional pathophysiological insights. However, we face the following objective constraints. To our knowledge, no large‑scale GWAS has yet focused on diabetes duration. Although GWAS data are available for preeclampsia and for type 1 and type 2 diabetes, diabetes duration—a time‑related variable—has not been systematically quantified or analyzed in these studies. Mendelian randomization analysis depends on published GWAS summary statistics; in the absence of GWAS data on diabetes duration, we cannot assess the causal effect of this temporal dimension within our current research framework. Instead, we indirectly approximated disease burden by analyzing markers of diabetes severity, such as glycated hemoglobin and fasting insulin levels. While these markers may correlate with disease duration to some extent, they do not fully capture its independent impact. We consider this a valuable research direction to be pursued in future work, especially once GWAS data specifically addressing diabetes duration become available.

6.Authors should mention inclusion and exclusion criteria regarding comorbidities which might interfere in their results.

Response:  Thanks for your valuable comments! Regarding potential comorbidities, our HbA1c and fasting insulin GWAS cohorts applied stringent exclusion criteria to minimize confounding: individuals diagnosed with type 1 or type 2 diabetes, those under diabetes treatment, and those meeting glycemic diagnostic thresholds (fasting glucose ≥7 mmol/L, 2‑hour glucose ≥11.1 mmol/L, or HbA1c ≥6.5%) were excluded. These stringent exclusion criteria ensured that the genetic association in the cohort was not affected by diabetes disease itself or its treatment, thereby providing genetic determinant of HbA1c and fasting insulin, and reduced potential confounding effects, which is critical for our causal inference. In addition, with regard to GWAS data for Type 1 Diabetes(T1D), Type 2 Diabetes (T2D), and preeclampsia, we utilized publicly available summary statistics derived from a case–control study design. As these data are at the summary level, further comorbidity screening was not feasible. We have acknowledged these limitations and discussed their implications for interpreting our findings.

Round 2

Reviewer 2 Report

Comments and Suggestions for Authors

well revision

Reviewer 3 Report

Comments and Suggestions for Authors

Many thanks to the authors, manuscript can be accepted for publication